# Copper Dysregulation in Major Depression: A Systematic Review and Meta-Analytic Evidence for a Putative Trait Marker

**DOI:** 10.3390/ijms26189247

**Published:** 2025-09-22

**Authors:** Rosanna Squitti, Mariacarla Ventriglia, Ilaria Simonelli, Cristian Bonvicini, Daniela Crescenti, Barbara Borroni, Mauro Rongioletti, Roberta Ghidoni

**Affiliations:** 1Molecular Markers Laboratory, IRCCS Istituto Centro San Giovanni di Dio Fatebenefratelli, 25125 Brescia, Italy; dcrescenti@fatebenefratelli.eu (D.C.); bborroni@fatebenefratelli.eu (B.B.); rghidoni@fatebenefratelli.eu (R.G.); 2Department of Laboratory Science, Research and Development Division, Ospedale Isola Tiberina-Gemelli Isola, 00186 Rome, Italy; maurociroantonio.rongioletti@fbf-isola.it; 3Department of Theoretical and Applied Sciences (DISTA), eCampus University, 22100 Como, Italy; 4Clinical Research Center, Ospedale Isola Tiberina-Gemelli Isola, 00186 Rome, Italy; mariacarla.ventriglia@fbf-isola.it (M.V.); ilaria.simonelli@fbf-isola.it (I.S.); 5Department of Biomedicine and Prevention, University of Rome Tor Vergata, 00133 Rome, Italy; 6Department of Clinical and Experimental Sciences, University of Brescia, 25100 Brescia, Italy

**Keywords:** major depression, copper, biomarkers, meta-analysis, females

## Abstract

Major depressive disorder (MDD) is a leading contributor to global disability. Despite advances in neurobiological research, no reliable peripheral biomarkers are currently available for diagnosis or monitoring. Copper (Cu), an essential trace element involved in redox balance and monoamine metabolism, has been repeatedly associated with MDD, though evidence remains inconsistent. To systematically evaluate and quantify differences in serum Cu concentrations between individuals with MDD and healthy controls, and to explore potential moderators, including sex, age, and analytical methodology. We conducted a systematic review and meta-analysis of observational studies reporting serum Cu levels in MDD patients versus controls. Data were extracted regarding diagnostic criteria, measurement methods, sample characteristics, and study quality. Subgroup and sensitivity analyses were performed based on demographic and methodological variables. Twenty-four studies, including 8617 participants (2736 MDD, 5881 controls), were analyzed. The pooled analysis revealed significantly higher Cu levels in MDD patients (Mean Difference (MD) = 2.22 µmol/L; 95% CI: 0.97–3.48; *p* = 0.001), although heterogeneity was high (I^2^ = 98.6%). Sub-analysis in females confirmed the association (MD = 1.39 µmol/L; 95% CI: 0.65–2.12; *p* = 0.009). Results remained robust in sensitivity analyses. Begg’s test did not indicate possible publication bias. Our findings support an association between altered Cu homeostasis and MDD. Elevated Cu levels were observed in most studies, including among females and in subclinical cases, suggesting a potential role as a trait biomarker. Standardization in measurement and longitudinal designs is needed to confirm Cu’s clinical utility.

## 1. Introduction

Major depressive disorder (MDD) is one of the most common and disabling psychiatric conditions globally. According to the Global Burden of Disease (GBD) study, it consistently ranks among the top five causes of years lived with disability (YLDs) across age groups and regions [1]. In 2019 alone, depression accounted for approximately 46 million YLDs, reflecting its early onset, chronicity, and severe functional impact [1,2]. Beyond disability, MDD is linked to elevated premature mortality from both suicide and comorbid medical conditions. Affected individuals face an estimated 10–15 years of reduced life expectancy and a two- to four-fold increase in all-cause mortality, especially due to cardiovascular, metabolic, and inflammatory diseases [3,4].

Biologically, MDD involves dysregulation of neuroendocrine, immune and metabolic systems, and metal dyshomeostasis, which contribute to psychiatric symptoms and accelerate systemic aging and somatic risk [5]. Given its prevalence and systemic impact, MDD is a key target for prevention and personalized care. Yet, diagnosis and monitoring still rely on clinical observation. There remains a pressing need for reliable peripheral biomarkers to aid in early detection, risk stratification, disease pathway identification, and treatment response monitoring in psychiatric disorders [6].

Copper (Cu), an essential trace element, plays a critical role in oxidative stress regulation, mitochondrial function, and neurotransmitter metabolism. It acts as a cofactor for enzymes such as dopamine β-hydroxylase, monoamine oxidase, and superoxide dismutase [7,8,9]. In the bloodstream, Cu exists both bound to ceruloplasmin and in a loosely bound or unbound state known as ‘non-ceruloplasmin-bound’ or ‘free Cu’. This labile fraction is biologically active and potentially toxic, as it can catalyze redox reactions. The copper-to-zinc (Cu/Zn) ratio has also been proposed as an indicator of oxidative and inflammatory imbalance. Dysregulation of Cu homeostasis—especially an increase in “free” Cu—has been linked to increased oxidative damage, glutamatergic dysfunction, and inflammation, which are core processes implicated in MDD pathophysiology [10,11,12].

Multiple observational studies have investigated serum Cu concentrations in MDD patients compared to healthy controls, yielding inconsistent findings. Several studies reported elevated Cu levels in MDD patients, suggesting a potential association between hypercupremia and depressive symptomatology [10,13,14,15,16,17,18]. For instance, Alghadir et al. observed significantly increased copper levels in younger individuals with depressive symptoms [10] (Children’s Depression Inventory [CDI] ≥ 13), across multiple age strata, while Islam et al. found elevated serum copper in patients diagnosed with MDD according to DSM-IV criteria [14]. Similarly, postmenopausal women with mild-to-moderate depressive symptoms also exhibited higher copper levels than controls [11], and Liu et al. [16], using a large Chinese cohort (n = 429 MDD vs. 4418 controls), found elevated Cu in subjects with depression symptoms based on the nine-item Patient Health Questionnaire (PHQ-9). Conversely, other studies found no significant differences or showed mixed results depending on subgroups or outcome metrics [11,19,20,21], and a few studies have reported decreased Cu levels in MDD patients, particularly among males or during acute depressive episodes [22,23].

Discrepancies across studies may reflect methodological variability [e.g., Inductively Coupled Plasma Mass Spectrometry (ICP-MS) vs. Atomic Absorption Spectroscopy (AAS) vs. colorimetric methods], sample characteristics (e.g., menopausal status, age, sex ratio), or biological matrices (e.g., total vs. free Cu). The lack of standardization in reporting units and analytical protocols further complicates direct comparisons.

A meta-analysis published by Ni et al. in 2018 [12], which included 21 studies, found a significant elevation in serum Cu among MDD patients compared to controls. However, that analysis had notable limitations, including high heterogeneity, lack of subgroup analysis, and omission of recent studies applying more refined analytical techniques and sex-specific analyses.

More recent evidence has emphasized the role of non-ceruloplasmin Cu (also known as “free Cu”) [24] and Cu/Zn imbalance [25], particularly in women with mood spectrum disorders, including bipolar and MDD supporting previous evidence, which suggested that increased Cu levels may persist even after clinical remission, pointing to a potential role as a trait marker [13].

Given the absence of clinically validated biomarkers for depression, an updated meta-analysis is warranted. This study aims to (i) quantify the difference in serum Cu levels between MDD patients and healthy controls; (ii) evaluate potential moderating factors such as age and sex; (iii) explore the relevance of Cu homeostasis in the search for clinically useful biomarkers in psychiatry.

## 2. Methods

### 2.1. Prisma Guidelines

This systematic review and meta-analysis were conducted in accordance with the 2020 Preferred Reporting Items for Systematic Reviews and Meta-Analyses (PRISMA) guidelines [26] (For more information, visit: http://www.prisma-statement.org/) (1 August 2025). The checklist table has been included as Appendix A. No ethics approval was needed for this study as the data came from published studies.

### 2.2. Eligibility Criteria

Studies were considered eligible if they assessed circulating Cu levels in MDD patients and in non-depressed control groups, regardless of participants’ age, sex, or ethnicity. Only original research articles published in peer-reviewed journals were included.

Both case–control and cross-sectional designs were accepted, provided that they reported quantitative Cu values in both groups. Studies presenting subgroup analyses (e.g., by sex or disease severity) were also included if sufficient data could be extracted or reconstructed.

### 2.3. Information Sources and Search Strategy

The search strategy was developed according to the PICOS framework, structured around the research question: “What are serum Cu levels in individuals with major depressive disorder (MDD) compared to healthy controls?”. The search terms included the following:

“major depressive disorder”, “depression”, “MDD”, and their related MeSH terms, combined with “copper”, “serum copper”, and “trace elements”, both as keywords and indexed terms. Boolean operators and truncations were applied where appropriate.

Database searches were conducted across PubMed, Scopus, and Web of Science, covering the period from January 1989 to April 2025.

Additional relevant studies were identified by screening bibliographies of articles identified through the search strategy. Searches in Google Scholar and preprint servers (bioRxiv, medRxiv) were also performed to detect potentially eligible manuscripts not captured in indexed databases.

### 2.4. Selection of Studies

The study selection process was independently conducted by two reviewers (MV, RS). Discrepancies were resolved through discussion and consensus; if consensus could not be achieved, the final decision was taken by a third researcher (CB). Initially, titles and abstracts of all individuated records were screened. The full texts of potentially eligible studies were then retrieved and assessed. When a full-text article was unavailable, the corresponding author was contacted to request it. Studies with overlapping populations were excluded, and only those providing the most comprehensive and up-to-date data were retained to avoid duplication. We report the screening and selection process using a PRISMA flow chart.

### 2.5. Data Extraction and Management

The primary outcome of interest was the concentration of serum Cu in individuals with MDD compared to healthy controls.

Two authors (MV, IS) independently extracted data from all eligible studies using a pre-structured Excel spreadsheet. Any disagreements were discussed until a consensus was reached.

For each included study, we extracted the following data: first author, year of publication and country; total number of participants in the MDD and control groups; demographic characteristics of participants (mean age, sex distribution, ethnicity when available); diagnostic criteria for MDD; method used to measure serum Cu (e.g., atomic absorption spectroscopy, ICP-MS). About the outcome, mean and standard deviation (SD) or median, range, or interquartile range (IQR) were extracted. When necessary, Cu units were converted to a common metric (µmol/L) to allow for comparison across studies. When only median and IQR or Range were reported, mean and SD were estimated as per Wan et al. (2014) [27]. When data were reported separately for subgroups (e.g., males and females), they were combined to obtain a single pooled mean and SD, using the metaHelper package in R (v 1.0.0).

### 2.6. Assessment of Risk of Bias in Included Studies

Risk of bias for included observational studies was assessed using the Newcastle–Ottawa Scale (NOS), evaluating selection, comparability, and outcome domains. Studies were rated independently by two reviewers, and discrepancies were resolved by consensus. The score was converted to the Agency for Healthcare Research and Quality (AHRQ) standards based on the following thresholds:-Good quality with 3 or 4 stars in the selection domain AND 1 or 2 stars in the comparability domain AND 2 or 3 stars in the outcome/exposure domain;-Fair quality with 2 stars in the selection domain AND 1 or 2 stars in the comparability domain AND 2 or 3 stars in the outcome/exposure domain;-Poor quality with 0 or 1 star in the selection domain OR 0 stars in the comparability domain OR 0 or 1 stars in the outcome/exposure domain.

NOS scores were used in sensitivity analysis to interpret the robustness of findings but were not used to exclude studies from the main meta-analysis.

### 2.7. Effect Measures

Since the outcome variables were continuous, effect sizes were calculated as absolute mean difference (MD) in serum Cu concentrations between patients with depression and healthy controls. The pooled effect was reported as MD with corresponding 95% confidence intervals (CI). A positive MD indicates higher Cu levels in depressed patients compared to controls, whereas a negative MD reflects lower Cu levels in the depressed group.

### 2.8. Data Synthesis and Statistical Analysis

Meta-analysis was performed using a random-effects model to calculate the pooled MD in serum Cu levels, accounting for between-study heterogeneity. Results of the primary analysis were graphically represented using forest plots.

Statistical heterogeneity was assessed using the Chi-squared (χ^2^) test and quantified by the between-study variance (τ^2^) and the inconsistency index (I^2^), each with its corresponding 95% CI. The I^2^ statistic describes the percentage of total variation across studies that is due to heterogeneity rather than chance, ranging from 0% (no heterogeneity) to 100% (maximum heterogeneity). Interpretation followed Higgins et al. (2003) [28], where I^2^ values of 25%, 50%, and 75% correspond to low, moderate, and high heterogeneity, respectively.

To explore potential sources of heterogeneity, meta-regression analyses were conducted to assess whether the proportion of female participants or the mean age in the depressed group influenced the magnitude of Cu level differences.

A *p* value less than 0.05 was considered statistically significant. All statistical analyses were performed using meta package (version 8.1-0) in R version 4.3.3.

### 2.9. Sensitivity Analyses

Sensitivity analyses were conducted to test the robustness of the findings. In particular, the meta-analysis was repeated after excluding studies that enrolled adolescent or school-aged participants (age < 18 years) to evaluate the influence of younger samples on the pooled effect estimate.

### 2.10. Publication Bias

Publication bias was assessed through visual inspection of funnel plots, where the SE was plotted on a reversed scale to position larger, more precise studies toward the top. The contour-enhanced funnel plot was also generated to inspect the relation of possible asymmetry patterns with the statistical significance, considering desired significance thresholds *p* < 0.1, <0.05 and <0.01. Begg’s test was applied if the meta-analysis included at least 10 studies.

## 3. Results

### 3.1. Selection and Characteristics of the Studies

The selection process is illustrated in the PRISMA 2020 flow diagram (Figure 1).

A total of 24 studies were included in the meta-analysis, encompassing 8617 participants: 2736 patients with MDD and 5881 healthy controls (Table 1). The average age of MDD patients was 38.6 years (range: 7–60 years), while healthy subjects had a mean age of 38.8 years (range: 7–58 years). On average, 60.4% of the MDD group were female (range: 25.7–100%), compared to 55.5% in the control group (range: 23.4–100%).

Two studies used the Chinese Classification of Mental Disorders, Second Revision (CCMD-2R) for diagnosis [18,29], while two others employed the Chinese Classification of Mental Disorders, Third Revision (CCMD-3) in combination with the Hamilton Depression Rating Scale (HDRS) [30,31]. One study used both the CCMD-3 and the International Classification of Diseases, 10th Revision (ICD-10) criteria [17]. The Children’s Depression Inventory (CDI) was used in one study [10], and the Diagnostic and Statistical Manual of Mental Disorders, Fourth Edition (DSM-IV) was used in two studies [11,23]. Three studies applied the Hamilton Depression Rating Scale (HDRS) in combination with the Young Mania Rating Scale (YMRS) [15,19,32], while two studies used the HDRS alone [13,33].

With reference to the method used for the analysis of Cu, most of the studies have used atomic absorption or its variants (78%; AAS 14 studies, 2 studies FAAS and one study EAAS, Table 1).

**Table 1 ijms-26-09247-t001:** Characteristics of included studies that reported serum Cu level in MDD and healthy controls.

Study	Country	Sample Size		Gender F (%)		Age [Years (SD)]		Cu [µmol/L (SD)]		Diagnostic Criteria	Biological Matrix	Method
Author (Year)		MDD	CTRL	MDD	CTRL	MDD	CTRL	MDD	CTRL			
Manser 1989 [34]	Pakistan	31	62	51.6	46.8	NA	NA	17.9 (3.7)	14.7 (1.8)	NA	whole blood	AAS
Narang 1991 [6]	Indian	35	35	40	40	matched	matched	19.2 (4.5)	16.8 (2.6)	HDRS	plasma	AAS
Maes 1997 [20]	Belgium	31	15	45.20	33	51.4 (13.5)	47.5 (15)	18.6 (4.7)	18.7 (1.9)	HDRS	serum	AAS
Yu 1997 [18]	China	22	26	63.6	NA	27.6 (9.2)	29.4 (8.7)	15.5 (4.7)	11.8 (2.3)	CCMD-2R	serum	AAS
Fernandez-Gonzales 1998 [23]	Spain	24	33	87.5	60.6	NA	NA	16.4 (3.8)	21.1 (6.1)	DSM-IV	serum	AAS
Schlegel-Zawadzka 1999 [13]	Poland	19	16	63.2	37.5	42.2 (10.6)	37 (9.1)	18.1 (2.7)	14.9 (1.4)	HDRS	serum	AAS
Chang 2001 [29]	China	68	66	51.5	50.0	32.0 (11.6)	31.5 (10.5)	19.0 (2.1)	17.8 (2.8)	CCMD-2R	serum	DCP-AES
Ma 2006 [30]	China	60	40	73.3	55	39.8 (14.2)	36.4 (2.1)	17.0 (2.6)	15.8 (1.7)	CCMD-3/HAMD	serum	AAS
Crayton 2007 [11]	USA	813	54	59.7	51.8	30–60	45.7 (7.0)	15.3 (4.6)	16.1 (2.7)	DSM-IV	serum	AAS
Wan 2008 [31]	China	70	64	25.7	23.4	23.2 (8.4)	30.9 (9.2)	22.2 (1.0)	16.3 (2.0)	CCMD-3/HAMD	blood	Polarography
Liu 2008 [17]	China	41	21	58.5	57.1	35.2 (12.8)	36.8 (11.3)	19.8 (3.3)	17.8 (2.8)	CCMD-3/ICD-10	serum	DCP-AES
Salustri 2010 [35]	Italy	13	13	84.6	84.6	54.2 (13.5)	55.9 (19.3)	16.5 (4.3)	13.5 (4.1)	DSM-IV/MADRS	serum	Colorimetry
Li 2014 [33]	China	65	65	55.4	53.8	38.5 (7.5)	38.7 (7.1)	20.1 (1.7)	15.3 (1.2)	HDRS	serum	Polarography
Alghadir 2015 [10]	Egypt	73	77	45.2	35.1	7–18	7–18	25.6 (3.4)	19.7 (2.5)	CDI	serum	AAS
Styczen 2016 [21]	Poland	114	50	75.4	72	49.4 (10.7)	45.8 (12.4)	12.7 (4.2)	14.3 (6.1)	MADRS-HDRS	serum	ETAAS
Skzup 2017 [36]	Poland	70	128	100	100	56.3 (5.5)	56.3 (5.6)	18.2 (1.2)	16.8 (3.5)	Beck	serum	AAS
Islam 2018 [14]	Bangladesh	247	248	63	59	33.0 (0.7)	33.5 (0.6)	21.9 (0.5)	15.9 (0.3)	DSM-IV	serum	AAS
Al-Fartusie 2019 [37]	Iraq	60	60	NA	NA	40–60	39–58	24.4 (1.8)	17.6 (2.1)	NA	serum	FAAS
Tanvir 2020 [38]	Pakistan	185	185	55.7	64.9	37.75 (11.5)	39.4 (12.6)	19.1 (6.2)	15.7 (4.2)	ICD10	serum	FAAS
Liao 2021 [15]	China	41	41	63.4	51.2	28.07 (10.1)	26.5 (7.9)	16.9 (4.7)	13.7 (2.8)	HDRS-YMRS	serum	AAS
Fu 2023 [22]	China	72	75	50	49	39.3 (15.5)	41.9 (6.9)	15.6 (1.8)	20.2 (3.3)	HAMD	serum	ICP-MS
Liu 2024 [16]	China	429	4418	59.4	49.0	49.47 (17.2)	47.4 (18.6)	19.6 (4.5)	18.6 (4.5)	PHQ-9	serum	ICP-DRC-MS
Zhong 2024 [19]	China	108	44	56.5	50	25.85 (7.4)	25.9 (3.8)	12.1 (2.3)	13.5 (2.8)	HDRS-YMRS	serum	AAS
Abd Rab El Rasool & Farghal 2024 [32]	Egypt	45	45	71.1	66.7	31.1 (5.5)	28.3 (6.3)	19.7 (6.1)	15.3 (2.2)	HDRS-YMRS	serum	AAS

AAS: Atomic Absorption Spectroscopy; CCMD-2R: Chinese Classification of Mental Disorders, Second Revision; CCMD-3: Chinese Classification of Mental Disorders, Third Revision; ICP-DRC-MS: Inductively Coupled Plasma-Dynamic Reaction Cell Mass Spectrometry; CDI: Children’s Depression Inventory; CTRL: controls; DCP-AES: Direct Current Plasma Atomic Emission Spectroscopy; DSM-IV: Diagnostic and Statistical Manual of Mental Disorders, Fourth Edition; DSM-V: Diagnostic and Statistical Manual of Mental Disorders, Fifth Edition; ETAAS: Electro Thermal Atomic Absorption; FAAS: Flame Atomic Absorption Spectroscopy; HAMD: Hamilton Depression Scale; HDRS: Hamilton Depression Rating Scale; ICD-10: International Classification of Diseases, 10th Revision; ICP-MS: Inductively Coupled Plasma Mass Spectrometry; MADRS: Montgomery–Åsberg Depression Rating Scale; MDD: Major Depressive Disorder; NA: not available; PHQ-9: nine-item Patient Health Questionnaire; YMRS: Young Mania Rating Scale. *30% of the samples were also confirmed using the AAS method.

### 3.2. Overall Analysis

The pooled analysis showed a significant difference in Cu mean levels between the two groups, with MD = 2.22 µmol/L (95% CI: 0.97; 3.48 µmol/L; *p* = 0.001) indicating higher level of Cu in MDD subjects but the heterogeneity was very high (I^2^ = 98.6%, 95% CI: 98.3–98.8%, *p* < 0.001; τ^2^ = 8.31, 95% CI: 4.82–17.01). In particular, three studies reported significantly lower Cu mean values in MDD patients compared to those of healthy controls (Figure 2).

A qualitative analysis of the studies considered evaluating quality by NOS is shown in Table 2.

Results from the meta-regression analysis, evaluating the percentage of females in the MDD group as a potential predictor of the MD in serum Cu levels, showed no significant effect (beta = −0.05, 95% CI: −0.13 to 0.02; *p* = 0.163). Similarly, meta-regression assessing the impact of the mean age difference between patients and controls revealed no significant association with the estimated MD in Cu levels (beta = −0.23, 95% CI: −0.67 to 0.20; *p* = 0.275).

Sensitivity analysis excluding studies with a female proportion greater than 80% in the MDD group [35,36] did not substantially alter the pooled estimate (MD = 2.23, 95% CI: 0.86 to 3.61; *p* = 0.003). The findings remained robust after excluding studies that included participants under 18 years of age [10,34], yielding a pooled MD of 2.00 (95% CI: 0.67 to 3.33; *p* = 0.005). Results also remained consistent after excluding studies rated as “poor” according to the Agency for Healthcare Research and Quality (AHRQ) quality standards (MD = 2.26, 95% CI: 0.88 to 3.63; *p* = 0.003; Table 3).

To further investigate the sources of the high heterogeneity observed in the overall analysis (I^2^ = 98.6%), we conducted sensitivity analyses by stratifying studies based on the method used to measure serum Cu. In particular, we analyzed a subgroup of 17 studies that employed atomic absorption spectroscopy (AAS) as their primary analytical technique. The pooled MD in Cu levels between MDD patients and healthy controls in this subgroup was 2.37 µmol/L (95% CI: 0.89 to 3.85; *p* = 0.004), indicating higher serum Cu in the MDD group. However, heterogeneity remained high within this subgroup, with I^2^ = 97.9% and τ^2^ = 7.50. To examine whether the biological matrix used for copper quantification contributed to the observed heterogeneity, we performed a sensitivity analysis stratifying studies by sample type. Most studies (n = 21) measured copper in serum, while a smaller subset employed plasma or whole blood (n = 3). In the serum-only subgroup, the pooled MD between MDD patients and healthy controls was 1.98 µmol/L (95% CI: 0.62 to 3.34; *p* = 0.004), with very high residual heterogeneity (I^2^ = 98.7%, τ^2^ = 9.60). In the plasma or whole-blood subgroup, the MD was even higher at 3.93 µmol/L (95% CI: 1.60 to 6.27; *p* = 0.001), but heterogeneity remained substantial (I^2^ = 91.8%, τ^2^ = 3.84). Furthermore, to explore whether the publication year contributed to heterogeneity in effect sizes, we conducted a sensitivity analysis stratifying studies into two groups: those published in or before 2010 (N = 12) and those published after 2010 (N = 12). The pooled MD in copper levels for studies published in or before 2010 was 1.87 µmol/L (95% CI: 0.36 to 3.38; *p* = 0.02), with substantial heterogeneity (I^2^ = 95.0%, τ^2^ = 6.42). For studies published after 2010, the pooled MD was higher, at 2.49 µmol/L (95% CI: 0.84 to 4.15; *p* = 0.003), but heterogeneity remained very high (I^2^ = 99.3%, τ^2^ = 8.87). We did not conduct a country-based subgroup analysis because this variable mainly indicates the country of publication rather than ethnicity, which may be more relevant.

### 3.3. Publication Bias

The funnel plot (Figure 3A) did not reveal substantial asymmetry suggestive of publication bias. The contour-enhanced funnel plot (Figure 3B) showed that most large, high-precision studies (i.e., with low SE) were located toward the top of the plot, displaying both significant (light gray area, *p* < 0.01) and non-significant (white area, *p* > 0.10) differences in mean Cu levels between MDD patients and healthy controls. Among smaller, less precise studies toward the bottom of the plot, results were distributed across both significant (light gray area, *p* < 0.01) and marginally significant regions (dark gray area, 0.01 < *p* < 0.05), suggesting no evident directional bias.

This visual interpretation was supported by the Begg test, which showed no significant evidence of publication bias (bias = 18, SE = 40.31; *p* = 0.655).

### 3.4. Meta-Analysis in Female Subjects

We conducted a subgroup meta-analysis including only female participants. Four studies were analyzed—Salustri et al., 2010 [35], Szkup et al., 2017 [36], Crayton et al., 2007 [11], and Zhong et al., 2024 [19]—encompassing a total of 820 individuals (629 females with MDD and 191 non-depressed controls).

The pooled MD was significantly positive (MD = 1.39 µmol/L; 95% CI: 0.65 to 2.12; *p* = 0.009), indicating that, even when restricting the analysis to female participants, serum Cu levels were higher in patients with MDD compared to healthy controls.

## 4. Discussion

This meta-analysis aimed to systematically evaluate differences in serum Cu levels between patients with MDD and healthy controls. Based on the currently reviewed studies, there is substantial evidence suggesting that serum Cu concentrations are elevated in individuals with MDD, although heterogeneity across studies remains notable. To further explore sources of heterogeneity, we performed sensitivity analyses stratifying the studies by copper assessment method, publication year, and biological matrix. While the heterogeneity remained high across most subgroup comparisons, restricting the analysis to studies using plasma or whole blood samples led to a modest reduction in heterogeneity (I^2^ = 91.8%), suggesting that biological matrix may partially account for the observed variability.

Several investigations—ranging from early clinical studies to recent high-resolution biochemical analyses—reported significantly higher Cu levels in MDD patients than in controls. This trend is consistent with the findings of the earlier meta-analysis by Ni et al. (2018) [12], which showed a pooled increase in serum Cu in MDD, albeit with substantial heterogeneity and limited subgroup analyses. Our inclusion of additional recent studies with improved methodologies and sample characterization (e.g., age and sex-specific analyses) adds clarity to this observation. However, not all studies concur. Some investigations found no significant difference in Cu levels between MDD patients and controls [19,20], while others even reported lower Cu levels, particularly in male patients [21,22,23]. This divergence may reflect variations in the form of copper measured (total vs. “free” Cu, also called non-ceruloplasmin Cu), and patient characteristics (e.g., sex, menopausal status, medication use). For instance, Styczeń et al. [21] found no significant differences in serum Cu levels between MDD subtypes and healthy controls. In contrast, Zhong et al. [19] reported lower Cu levels specifically in male MDD patients, along with an elevated Cu-to-zinc (Cu/Zn) ratio and neurometabolic alterations in the anterior cingulate cortex. These included a reduction in the N-acetyl aspartate to creatine ratio (NAA/Cr), a recognized marker of neuronal integrity measured via proton magnetic resonance spectroscopy (^1^H-MRS). Together, these findings suggest that Cu dysregulation may influence brain function in a sex-specific manner in MDD, possibly through mechanisms involving oxidative stress and glutamatergic neurotransmission.

Mechanistically, elevated Cu may exacerbate oxidative stress via Fenton-like reactions and promote neuroinflammation through dysregulated ceruloplasmin and cytokine signaling [39,40]. In particular, “free” Cu has been linked to neuronal excitability and glutamate toxicity [24,35]. Salustri et al. demonstrated that increased free copper in MDD patients correlated with altered cortical excitability, suggesting a direct functional role [35]. Similarly, Squitti et al. [24] identified altered Cu parameters in both unipolar and bipolar depression, further reinforcing Cu’s transdiagnostic relevance. However, this evidence remains limited: only two studies to date—Salustri et al. (2010) [35] and Squitti et al. (2024) [24]—have explored free copper in mood disorders, with the latter assessing a transdiagnostic population without separate analysis for MDD. As such, our meta-analysis focused on total copper values, which are more widely available. Future studies should prioritize systematic evaluation of free copper in well-characterized MDD cohorts, to further elucidate its trait-like biomarker potential.

Interestingly, Schlegel-Zawadzka et al. [13] reported that copper levels remained elevated in MDD patients even after clinical remission, suggesting that Cu may function as a trait marker rather than merely reflecting the current disease state. This raises the possibility that Cu imbalance could represent an endophenotype of depressive vulnerability, potentially offering predictive or diagnostic utility. The hypothesis that Cu dysregulation may represent a trait-like biological alteration in mood disorders is supported by findings from studies showing elevated Cu levels irrespective of clinical state or severity. For instance, Squitti et al. [25] found that increased serum Cu and altered Cu/Zn ratios significantly distinguished psychiatric patients from healthy controls, independently of symptom severity scores. Similarly, Alghadir et al. [10] reported elevated Cu concentrations in schoolchildren with subclinical depressive symptoms, suggesting that Cu imbalance may emerge early and precede formal diagnosis. These observations reinforce the possibility that Cu may act as a putative trait biomarker in affective disorders [13].Although there are promising associations, caution must be exercised. Most studies are observational and cross-sectional, making causal inference impossible. The heterogeneity noted in most studies is not predominantly qualitative but is largely due to variations in study execution and sample size. However, it is important not to overlook the qualitative heterogeneity evident in the main analysis. Additionally, even among studies that report a negative mean difference, a significant variance in sample size is apparent. Funnel plots and statistical tests such as Begg and Egger are unreliable for detecting publication bias when there is high between-study heterogeneity or small study effects, since skewness in such cases may stem from actual differences between studies rather than systematic bias, and statistical tests may lack power and provide misleading outcomes. Examining subgroup effects and sources of heterogeneity can be difficult, with excessive reliance on post hoc subgroup or sensitivity analyses being a notable issue [41]. Additionally, only some studies account for confounding factors like diet, inflammation, and medication use. Variability in measurement methods (e.g., AAS versus colorimetric assays) and inconsistent reporting of ceruloplasmin-bound versus free Cu are further limitations. These methodological challenges likely contribute to the heterogeneity observed and restrict the capacity to draw firm conclusions about the clinical usefulness of Cu as a biomarker. Nonetheless, significant heterogeneity in a meta-analysis is always important to note. It may reflect high clinical variability among included studies, suggesting that estimating an overall effect may not be appropriate. Conversely, thoroughly exploring heterogeneity might uncover poor study design or, fail to identify the cause; in both cases, investigating the origins of heterogeneity could be a focus for future research [42]. Future investigations should aim to incorporate comprehensive clinical, nutritional, and biochemical profiles, including ceruloplasmin levels and inflammatory markers, to better clarify the role of Cu homeostasis in MDD.

## 5. Conclusions

The present findings support the hypothesis that Cu dysregulation is implicated in MDD, and may reflect underlying pathophysiological mechanisms involving oxidative stress, neuroinflammation, and mitochondrial dysfunction. Our findings suggest that copper dysregulation may hold potential as a trait biomarker in MDD; however, further validation is warranted through longitudinal and mechanistically oriented studies. Specifically, future research should prioritize prospective longitudinal designs, standardized quantification of free vs. total Cu, and integration of biochemical, neuroimaging, and genetic data.

Identifying reproducible, clinically accessible biomarkers like Cu may contribute to a biologically informed classification of depression, aligned with initiatives such as the Research Domain Criteria (RDoC) framework [43].

## Figures and Tables

**Figure 1 ijms-26-09247-f001:**
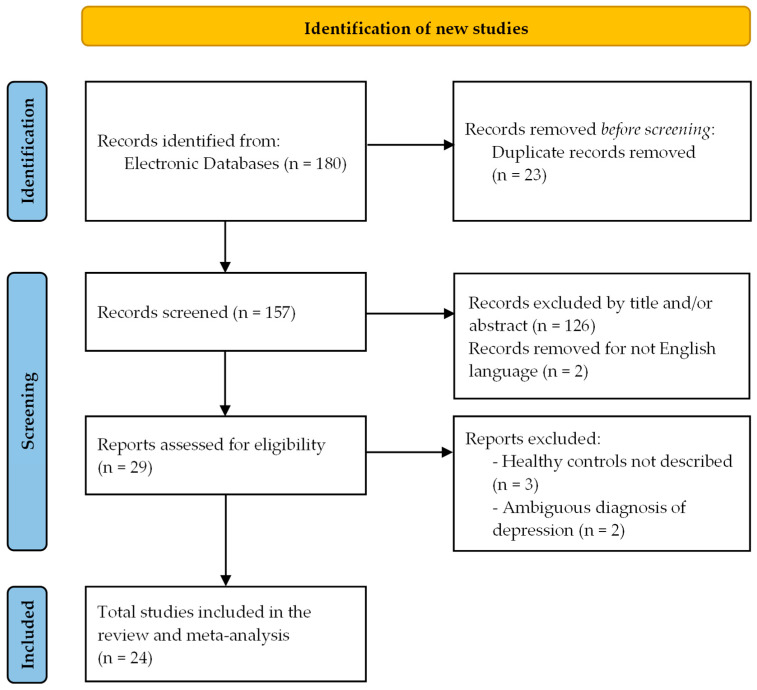
PRISMA 2020 flow diagram of the study selection procedure for systematic reviews, which included searches of databases.

**Figure 2 ijms-26-09247-f002:**
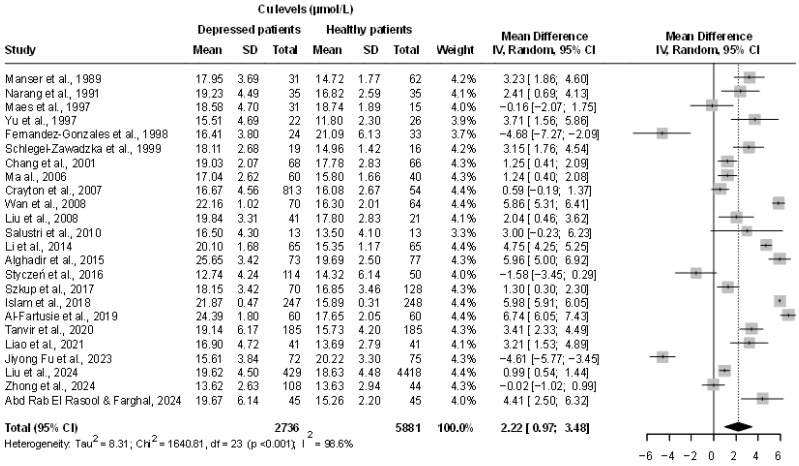
Forest plot of the meta-analysis comparing serum Cu levels between depressed patients and healthy controls: Each study is represented with its mean difference (MD) and 95% confidence interval (CI). A positive MD indicates higher Cu levels in patients with depression. The pooled effect size, shown as a diamond, suggests significantly elevated Cu in the depressed group. Box sizes reflect study weights. Considerable heterogeneity was observed across studies (I^2^ = 99%) [6,10,11,13,14,15,16,17,18,19,20,21,22,23,29,30,31,32,33,34,35,36,37,38].

**Figure 3 ijms-26-09247-f003:**
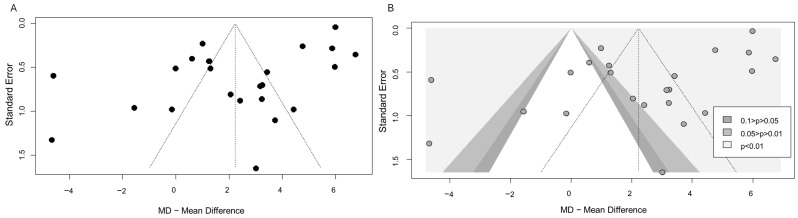
Funnel plot panel (**A**) and contour-enhanced funnel plot panel (**B**) illustrating potential publication bias across studies included in the meta-analysis. Each dot represents an individual study. In panel (**A**), the plot appears symmetric, suggesting no major publication bias. In panel (**B**), the background shading indicates areas of statistical significance: white (*p* > 0.10), dark gray (0.01 < *p* < 0.05), and light gray (*p* < 0.01). Most large studies cluster at the top of the funnel with both significant and non-significant results. Smaller studies are more dispersed but are distributed across both significant and non-significant regions, supporting the absence of directional bias.

**Table 2 ijms-26-09247-t002:** Quality assessment of studies included in the meta-analysis.

Study	Adequate Definition of Cases	Representativeness of Cases	Selection of Controls	Definition of Controls	Comparability	Exposure Assessment	Same Method	Non-Response Rate	Total Quality Scores	AHRQ Standards
Manser 1989 [34]	0	0	1	1	1	1	1	1	6	Fair
Narang 1991 [6]	1	0	1	1	1	1	1	1	7	Good
Maes 1997 [20]	1	1	1	1	2	1	1	1	9	Good
Yu 1997 [18]	1	1	1	1	1	1	1	1	8	Good
Fernandez-Gonzales 1998 [23]	1	1	0	1	2	1	1	1	8	Good
Schlegel-Zawadzka 1999 [13]	1	0	0	1	0	1	1	1	5	Poor
Chang 2001 [29]	1	1	0	0	1	1	1	1	6	Fair
Ma 2006 [30]	1	1	1	1	1	1	1	1	8	Good
Crayton 2007 [11]	0	0	0	1	1	1	1	1	5	Poor
Wan 2008 [31]	1	0	1	1	2	1	1	1	8	Good
Liu 2008 [17]	1	0	0	1	1	1	1	1	6	Fair
Salustri 2010 [35]	1	0	1	1	1	1	1	1	7	Good
Li 2014 [33]	1	0	0	1	1	1	1	1	6	Fair
Alghadir 2015 [10]	1	0	1	1	2	1	1	1	8	Good
Styczeń 2016 [21]	1	1	0	1	2	1	1	1	8	Good
Szkup 2017 [36]	1	0	1	1	2	1	1	1	8	Good
Islam 2018 [14]	1	1	1	1	1	1	1	1	8	Good
Al-Fartusie 2019 [37]	0	1	1	0	1	1	1	1	6	Fair
Tanvir 2020 [38]	1	1	1	1	1	1	1	1	8	Good
Liao 2021 [15]	1	1	1	1	1	1	1	1	8	Good
Fu 2023 [22]	1	1	1	0	1	1	1	1	7	Good
Liu 2024 [16]	1	1	0	0	1	1	1	1	6	Fair
Zhong 2024 [19]	1	1	1	1	1	1	1	1	8	Good
Abd Rab El Rasool & Farghal 2024 [32]	1	1	1	0	1	1	1	1	7	Good

The score values were described in Section 2.6. “Assessment of risk of bias in included studies”.

**Table 3 ijms-26-09247-t003:** Sensitivity analysis results.

Sensitivity Analyses	Number of Studies	MD	95% CI	*p*	I^2^
Excluding Salustri et al., 2010 and Szkup et al., 2017 [35,36]	22	2.23	0.86–3.61	0.003	98.7%
Excluding Manser et al., 1989 and Alghadir et al., 2015 [10,34]	22	2.00	0.67–3.33	0.005	98.7%
Excluding studies with AHRQ standards equal to poor [11,13]	22	2.26	0.88–3.63	0.003	98.6%

MD, mean difference; CI, confidence interval; *p*, *p*-value; I^2^, inconsistency index.

## Data Availability

No new data were created or analyzed in this study. Data sharing is not applicable to this article.

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
