# Peer review of "Copper Dysregulation in Major Depression: A Systematic Review and Meta-Analytic Evidence for a Putative Trait Marker"

_ijms, 2025, doi:10.3390/ijms26189247_

Round 1
Reviewer 1 Report
Comments and Suggestions for Authors
The manuscript entitled “Copper Dysregulation in Major Depression: Meta-Analytic Evidence for a Putative Trait Marker” represents a meta-analysis, which is based on a final set of 24 studies aimed to identify a link between copper level in serum and liability to clinical depression. The study explains a relevance to perform a meta-analysis to make a conclusion on the use of peripheral biomarkers, such as copper levels for predicting depression-related outcomes. The authors carried out the present study in accordance with the PRISMA flowchart selecting studies from 1989 to 2025. This manuscript is a well-written article summarizing existing studies in the chosen field of research. The study has several advantages, including performing subgroup and sensitivity analysis, assessing the robustness of their findings, discussing the sources of possible heterogeneity and reporting limitations.
Several issues need to be addressed.
- In the Introduction I suggest to reorganize the paragraph (lines 66-81) with respect to a direction of copper effect on predisposing to MDD in previous findings, i.e., it would be more readable first to report the results of studies demonstrating, for example, the association of high Cu levels with depression followed by a reverse relationship.
- It remained unclear for me whether the authors have recalculated their results while deleting the studies that provided copper levels measured in plasma and whole blood?
- Please, start the names of the columns with capital letters in Table 2.
I can suggest to accept after minor revision, providing that the authors addressed my comments.
Reviewer 2 Report
Comments and Suggestions for Authors
This manuscript addresses an important and timely research question: whether serum Cu dysregulation is associated with MDD and whether it may serve as a trait biomarker. The authors have conducted a systematic review and meta analysis of 24 studies including 8617 participants, following PRISMA guidelines. The study is generally well designed and well written. However, several points require attention before the manuscript can be considered for publication.
1. Although acknowledged, the extreme heterogeneity substantially limits the interpretability of the pooled effect. I recommend that additional stratified analyses should be considered to better dissect sources of variability.
2. Most included studies measured total serum Cu; very few assessed non-ceruloplasmin or free Cu. The conclusions about trait biomarker potential would be stronger if analyses distinguished between total Cu and free Cu. Without this, the biomarker claim is somewhat overstated and should be restricted.
3. Several included studies were rated as “poor” or “fair.” While sensitivity analyses excluded poor-quality studies, the overall conclusions still rely heavily on heterogeneous and variable-quality evidence.
4. Although the funnel plot and Begg/Egger’s tests are reported, the narrative should acknowledge the limitations of these tests in the presence of high heterogeneity and small-study effects.
5. Many studies did not account for diet, ceruloplasmin, inflammation, comorbidities, or medication use. These confounders should be discussed as major limitations, since they may strongly influence serum Cu.
6. The conclusion states “a potential role as a trait biomarker.” It could be more cautious to state “Cu dysregulation shows potential as a trait biomarker, but further validation is required.”
7. The authors should clarify terms such as “free Cu,” “non-ceruloplasmin Cu,” and “Cu/Zn ratio” earlier in the introduction.
Round 2
Reviewer 2 Report
Comments and Suggestions for Authors
Well revised by the authors.